# Emotional Universe of Nurse Case Managers Regarding Care for Elderly at Risk in Spain: A Hermeneutical Study

**DOI:** 10.3390/ijerph192416445

**Published:** 2022-12-08

**Authors:** Ángeles Pastor-López, Pedro E. Ventura-Puertos, José Hernández-Ascanio, Vanesa Cantón-Habas, Pablo Martínez-Angulo, Manuel Rich-Ruiz

**Affiliations:** 1Instituto Maimónides de Investigación Biomédica de Córdoba (IMIBIC), Universidad de Córdoba (UCO), Hospital Universitario Reina Sofía (HURS), 14004 Córdoba, Spain; 2Distrito Sanitario Córdoba-Guadalquivir, Servicio Andaluz de Salud (SAS), 14011 Córdoba, Spain; 3Interdisciplinary Research Group in Discourse Analysis (HUM380), Universidad de Córdoba (UCO), 14071 Córdoba, Spain; 4Ciber Fragility and Healthy Aging (CIBERFES), 28029 Madrid, Spain; 5Nursing and Healthcare Research Unit (Investén-isciii), Instituto de Salud Carlos III, 28029 Madrid, Spain

**Keywords:** nurse case managers, emotions, hermeneutics, elderly at risk

## Abstract

The role of nurse case managers (NCM) involves a rarely visible emotional labor, even more when their role focuses on the care of elders at risk (EAR). Motivated by the lack of qualitative research on the emotional universe of NCM, this study explores the emotional universe (EU) of NCM regarding the care they provide to EAR in primary health care as well as the reasons that generate these emotions. An interpretative–phenomenological approach was implemented in southern Spain, with a purposive sampling that included nurses playing the NCM role for at least three years. Data collection was conducted in two periods (between September 2019 and July 2022). The primary collection tool was the semi-structured individual interview, with starting categories based on Bisquerra’s EU taxonomy. The analysis followed Ricoeur’s considerations, using the Nvivo software. In the NCM’s EU, the recognition of the social phenomena stands out, with an open feeling of empathy regarding the desire of the EAR to continue living at home. However, there was also helplessness, resignation, disappointment, and frustration when EAR rejected their proposals. Furthermore, the system’s limitations aroused compassion in the NCM and made them go beyond the limits of their role. This EU requires that their role be valued more, and higher responsiveness must be enforced to improve EAR care.

## 1. Introduction

The role and collaborative process of nurse case managers (NCM) consist of the following five constitutive actions to meet health needs, as described in the scientific literature: to assess, plan, implement, coordinate, and value options and services [1]. This role involves rarely visible emotional phenomena, such as emotional labor, well-being, satisfaction, and inspiration that can derive from it [2,3,4]. This invisibility of their experience and affective impact is even larger when their work focuses on the complex care of older patients, in general, and those considered “elders at risk” (EAR, defined below) in particular.

Undoubtedly, the increase in the life expectancy of the population has represented an important social achievement globally, with the estimation that by 2050, people over the age of 65 will constitute 16% of the population [5]. In countries such as Spain, this percentage reached—already in 2019—19% of the total population, with 3% over 85 years, and an exponential growth is expected for the next few decades [6].

In the Spanish context, in a few decades, this country has positioned itself in the world’s leading group in life expectancy at birth, and by 2040, it could lead it, ousting Japan from first place [7]. However, this increase in life expectancy leads to an increase in the number of chronic diseases or accidents occurring in the final years of life (in line with the rest of the industrialized countries). This results, sooner or later, in some type of disability that, in turn, will diminish personal autonomy, causing an increase in the demand for care, especially in long-term care [7,8]. In Spain, as in the rest of the leading European countries, most chronic diseases fall on the elderly [7,8]. Furthermore, the most significant health-care expenditure incurred by individuals is concentrated in the last years of their lives; the older population naturally (or automatically) becomes the greatest beneficiary of public health care [7].

Currently, the Primary Health Care Systems (PHC) provide this care mainly in the home of the elderly [9,10]. The PHC prioritizes the so-called EAR patients. These are identified as dependent elderly—or at risk of dependence—older than 80. EAR patients may present some or several of the following characteristics: living alone, isolated or without a competent socio-family network, with housing and economic problems, possible victims of abuse, or at risk of premature or inadequate institutionalization [11].

Therefore, effective and efficient responses from various public administrations are required. Among these responses, there is the establishment of new nursing roles with a significant deepening and expansion of their competencies. Therefore, the emergence of the role of NCM is globally considered paradigmatic in this sense, as it is in the Spanish context [1,12].

Nurses are the professionals who mainly perform the role of Care Manager in PHC—instead of others, such as physicians. This fact is not strange, because the role of nurses has always placed them in a unique and integral perspective when approaching, observing, and closely accompanying individual and group vulnerability, especially in times of crisis, such as that generated by the COVID-19 pandemic [13,14,15]. Their competencies facilitate both the individualization and the contextualization of care, thus making possible the empowerment of patients [16] and the joint decision making when planning care [1]. The NCM have also become a reference figure in the area of PHC in Spain, planning and coordinating the care of people in situations of vulnerability and, especially, in the most complex cases, such as those presented by EAR [1].

In the context of EAR care, NCM and other PHC professionals are often overwhelmed [17,18,19]. The deterioration of the public health system and the precariousness of the care it provides—phenomena characteristic of the neoliberal environment in which Spain participates [20,21,22]—contribute to this situation. Specifically, the inefficiency of the social care system (such as those derived from the so-called Dependency Law [23]), in addition to the lack of support from the social–family network [24], overloads the health care system. The consequences of this overload on the emotional state of NCM seem to be expected, especially in times of pandemic. However, studies investigating this and the possible impact of this emotional cost on the EAR are scarce [25,26].

The most visible work of nurses is the provision of social and health care in situations of high vulnerability, which involves emotional labor [27] that is rarely recognized, even by the professionals who suffer it [28,29]. However, the consequences are not only worrying in the psycho-affective field—with high emotional suffering [30], compassion fatigue [31], and burnout [32]—but also in the physiological field, with cardiovascular manifestations, such as arterial hypertension or musculoskeletal type problems [33], and even in the professional development of nurses, undermining their work competence [34]. Moreover, this aspect is closely linked to emotions because the Emotional Determinants of Health are considered fundamental due to their influence on aspects such as job satisfaction, teamwork, leadership, or decision-making processes, among others [25,26,35,36,37,38].

These consequences, which have been widely described in the scientific literature when referring to nurses in general, have not been addressed concerning their affective experience or its determinants, especially in NCM.

Finally, and as regards the conceptual framework of this proposal, although the literature gathers different classifications, the taxonomy Universe of Emotions has been chosen [39] because of the visualization it offers of affective experiences as a strategy for their better understanding [40]. This taxonomy presents and orders numerous emotional labels, organizing its structure ‘‘around two great constellations’’: one of the positive emotions (Galaxies of Joy, Love, and Happiness) and the other of negative ones (Galaxies of Anger, Disgust, Fear, Anxiety, and Sadness). The Galaxy of Surprise occupies a central position, as do social emotions, such as shame and guilt [39]. This Universe of Emotions would allow us to comprehensively approach the emotional experiences of the NCM working with EAR, facilitating thus a view not explored to date of the work context of these professionals.

The present study aims to understand the Emotional Universe of NCM regarding the care they provide to EAR in PHC, interpreting their affective experiences related to their socio-work environment, to the preferences and decisions of patients, and to the existing limitations—both of the system and their own.

## 2. Materials and Methods

### 2.1. Design

This study is part of the thesis project of the lead author of this article, entitled: “Care of elders at social risk: Experiences of a complex social health reality” [41]. Nevertheless, it constitutes a different analysis due to the appearance, as an emerging result, of the affective experiences of the NCM that were part of its sample.

In this way, the study of the emotions existing in the collected data corpus was proposed in a particular way. The endemic phase of COVID-19 in Spain was added (as a second sociohistorical moment) to data existing before the emergence of the COVID-19 pandemic. The dates will be specified in the following sections.

An interpretative phenomenology [42] was designed to analyze the emotions of the NCM participating in the PhD thesis mentioned. In this sense, the use of hermeneutics has allowed a better understanding of what they are, that is, of their contextual references [43], at two specific historical and social moments [44,45].

Moreover, the understanding of ourselves—interpreters who use language as a tool [46] and are emotionally committed with our environment [47]— is intimately linked to our understanding of others [48], which is an aspect that also justifies the use of hermeneutics in this study.

Finally, it is worth noting the wide use of hermeneutics within the nursing discipline in dealing with the understanding and interpretation of the particularities of the care the nurses perform and the relationships and affections derived [47,48].

### 2.2. Environment

The study was conducted in a Spanish PHC district, specifically in a province in Andalusia, the most populated region of the country, located in the south of the Iberian Peninsula belonging to the so-called Cordoba-Guadalquivir Health District (DSCG). This district comprises two zones: Cordoba district, in the capital, comprising 13 NCM, and Guadalquivir district, comprising seven NCM. Between both districts, 461,078 inhabitants are served, from urban and rural areas, with an aging index of 121.253 in the Cordoba District and 119.733 in the Guadalquivir District [49].

The sampling was purposive and had the intensity [50] strategy to ensure the inclusion of individuals who had experienced the phenomenon studied in a lasting and significant manner. For this reason, the following inclusion criteria were considered: working currently as NCM for the Andalusian Health Service and having held this position for at least three years (this minimum of three years was based on Hill [51], who suggested that two to five years of nursing experience were needed to improve the quality of care). Priority was also given to NCM participating in the thesis project sample. However, the sampling was open to incorporating other NCM that were not included then.

Recruitment of the subjects was conducted by the lead author of the study; she was an NCM colleague of all of them and with whom she had a previous coworker relationship. At the beginning of the two periods, the author contacted each participant by telephone, explaining the purpose and asking them to participate.

### 2.3. Data Collection

The first data collection, corresponding to the thesis project mentioned above, was performed between September 2019 and February 2020, while the second data collection was performed between June and July 2022.

The main collection tool was the individual semi-structured interviews. Focus group interview was also used.

An interview script was drafted for both individual and group interviews. In the case of the pre-pandemic script, the starting categories based on the taxonomy of the Universe of Emotions were used [39]. In contrast, for the time of endemism, an interview script was drafted based on the results of a first analysis of the documentary data corpus consisting of the verbatim transcripts of the semi-structured interviews and discussion group conducted to the NCM corresponding to the thesis project (Pre-pandemic period) (Table 1).

All the individual interviews were conducted by the lead author. For the group interviews, one of the co-authors was moderator, while the lead author played the role of observer (both at the first and second periods). The duration of the individual interviews ranged from 40 to 80 min; the group interviews lasted approximately 90 min. All interviews were conducted face-to-face, seeking a suitable area, without interruptions or distractions, that was chosen by the interviewees. With these premises, the individual interviews were conducted in the NCM’s consulting rooms of the health-care centers of the DSCG except for one meeting that took place on-line because of the personal situation of the NCM interviewed. The Faculty of Medicine and Nursing of the University of Cordoba hosted both the first pre-pandemic group interview and the second group interview in one of their meeting rooms. All sessions were audio-recorded and subsequently transcribed by the lead author using the application “Word 2016” to facilitate their review and analysis.

To ensure the anonymization and confidentiality of participants, the coding of their names (from transcripts onwards) was performed as follows: (A) the acronym “NCM” would be used followed by a number of already assigned in the thesis data corpus, according to the order of interviews conducted; (B) for participants who were interviewed a second time, the suffix “POST” would be added to the acronym and number; (C) for those interviewed only in the endemic period, the (order) number should be added as a prefix to the set consisting of “NCMPOST”; (D) for the subjects of the pre-pandemic group, “NCMPRE” plus the order number of their first intervention would be chosen; € for those of the second group, “NCMPOST” plus the order number of their first intervention was chosen.

### 2.4. Data Analysis

Given the interpretative phenomenological nature of the study, the considerations proposed by Ricoeur were followed to conduct a hermeneutic analysis [52]. Nvivo qualitative analysis software [53] was used at all stages and levels. The analysis of the specific pre-pandemic corpus for this study began on 7 June and ended on 16 June 2022. The first interview to be analyzed was NCM5, and this was followed (in descending order of references from informants identified by Nvivo) by NCM6, NCM9, NCM4, and NCM8. Later, it was triangulated with the group interview of the same period. Then, following consensus from the research team, the script was created for the interviews (Table 1) to be conducted later. Once transcribed, the following were incorporated into the analysis: NCM9POST, 1NCMPOST, NCM6POST, 2NCMPOST, NCM5POST, and, finally, the group interview. The Universe of Emotions taxonomy constituted the starting category for the pre-pandemic corpus, whereas the emotions induced by its analysis—which constituted the basis for the questions in the semi-structured interview script and are listed in Table 1—were used for the time of endemism.

Concerning the iterative analytical process, in the preliminary analysis phase, the lead author thoroughly read the above-mentioned interview from NCM5, obtaining a general meaning, and wrote a list of ideas until reaching an intuitive explanatory framework. The analysis of her verbal and non-verbal narrative was then initiated, following the three levels proposed by Ricoeur [52].

At the first level (Explanation), nodes or units of complete meaning (meaning, explanation, and justification) related to the heading categories were selected. In doing so, the text was revised to verify that no meaningful meaning had been overlooked. Subsequently, 24 analytical subcategories linked to the starting ones were created.

The nodes were reconfigured at the second level (naïve Understanding) according to these subcategories. Once organized in subcategories, the nodes were regrouped by the affinity of meanings, creating codes capable of achieving a larger abstraction in their meaning level and discriminating the different characteristics of the same idea. Here, the rest of the pre-pandemic corpus began to be nodified in the order indicated, associating each new node with existing codes or configuring it in new encodings according to new meanings. Once the nodes of the rest of the interviews were incorporated (resulting in 12 main codes and 33 associated subcodes), those interviews conducted in 2022 were incorporated similarly.

Then, it was moved toward the third level (In-depth understanding), where the author reached a better understanding of the emotions of the NCM through the so-called “hermeneutic arc” [52]. The lead author considered both her pre-comprehension of the analyzed phenomenon and the rereading of the corpus of data, reaching the highest level of abstraction of meanings with the creation of patterns, which was derived into categories and subcategories.

### 2.5. Rigor

The criteria of rigor and quality followed in the investigation process were those established by Calderon [54]. Their summary is shown in Table 2.

### 2.6. Ethical Considerations

The study was conducted in compliance with the principles of the Declaration of Helsinki [55] and was approved (Minutes No. 277 Ref: 3924) by the Ethics Committee for the Province of Cordoba (Spain). Verbal and written consent was obtained by the lead author after providing verbal and written information to the participating NCM. The personal data obtained have been processed following the General Data Protection Regulation EU/2016/679, of 27 April 2016, and the provisions of the Organic Law 3/2018, of 5 December, on the Personal Data Protection and Digital Rights Guarantee.

## 3. Results

Discourse saturation in the pre-pandemic period was reached after analyzing the responses of 11 participants. To analyze the period after COVID-19, the same 11 participants were contacted. However, the life and contractual changes suffered by three of them (NCM4, NCM and NCM PRE2) meant that they were unable to participate in the second instance. In their place, three new participants were incorporated. Therefore, in the period after COVID-19, the discourses of 11 informants were collected. So, a total of 14 NCM (from 20 DSCG) participated in the study. Table 3 shows that seven NCM participated in the individual interviews and nine participated in the group interviews (two subjects participated in both types of interviews). Eleven participants were women and three were men; all participants had extensive previous experience in the role of case management in the primary care setting. Gender and years of experience have been considered, in the study, as criteria of variability/heterogeneity. However, the analysis of the participants’ discourse did not show conclusive differences

To generate the EU of NCM, these five themes are represented by visual elements (circular and progressive green patterns) that relate to the following insight: the experiences lived during COVID-19 have intensified the crisis context existing before its occurrence. First, there is the problematic life scenario of the elderly at risk, and then, there are the even more pronounced constraints and pressures to which NCM are subjected in the current circumstances of the precariousness of the social and health context (Figure 1).

These patterns describe different trajectories (according to their categories and subcategories) within the emotional galaxies of Bisquerra (Figure 2).

As Figure 2 shows, in the EU of NCM, several emotions are directly linked, such as sadness–compassion, compassion–pride, and perplexity–shame–indignation. These associations are contextualized in the narrative description of the results below. On the other hand, absences stand out in the figure, such as the emotions belonging to the galaxies of happiness, joy, and disgust, which, although present in the participants’ discourse, did not reach saturation. 

### 3.1. Recognizing “the Social Issues,” the Team, Women, the Rural Environment…

The group of NCM interviewed showed that in support of EAR care, social aspects took on a priority, especially after the impact of the pandemic:


*“When the social environment is more unfavorable, the process is more tortuous, less controlled, admissions, poor health, more dependency…”. *

*NCM6*



*“When they could not go outside, having gone outside less and so has generated a series of problems and all this has also brought a great deal of immobility and problems of loneliness and isolation”. *

*2NCMPOST*


With full awareness of the importance that the social phenomena acquire in their work, the NCM considered the team composed of nurses, doctors, and social workers as a fundamental bastion:


*“There are health teams, doctor and nurse, where they work very well for the approach they make of their homes, for the involvement that there is, for the professional responsibility.” *

*NCM8*



*“The fundamental bastion of these patients is the primary care team.” *

*NCM6*



*“My social worker is charming and a problem solver.” *

*1NCMPOST*


Moreover, they all recognized women—even patients—for their active role in care, contrasting with the limited role of men in care. They said that women, sooner or later, took the less friendly part of the care:


*“Women are always the ones getting the worst of it; moreover, they are the ones setting their shoulders to the wheel.” *

*NCM9*



*“In the approach to care, for example, I have seen that they go to the wall a bit more than usual; it is always unbalanced […]. If it was a 70/30 before, now it is more, a 90/10”. *

*NCM6POST*



*“The fragility is much greater when the caregiver is an elder man, in these conditions, than when the caregiver is a woman.” *

*NCMPOST2*


The last main recognition indicated here was that made by the NCM of the citizen and community environment, mainly the rural and neighborhood, for judging it close, supportive, and compassionate:


*“The ties are different, the... the support of the neighbors is much greater than, for example, in, in the city, in the city many times they do not know who lives next door.” *

*2NCMPOST*



*“’What’s up, Mary? How are you? How are you doing today?’ And that neighborhood conviviality, I believe it is greater in the town than in the city.” *

*NCM5*



*“And who is in charge? If the family, there is no family, or the children are in Germany, there is also the case of Germany, of Madrid... And then, in those cases, it is the neighbors who take care, at least in my area, in my area they are caring people.” *

*NCMPRE1*


### 3.2. Connecting with the Wish to Stay Home

Through empathy, they could understand how, despite different situations of vulnerability related to feelings of loneliness and neglect, elders want to continue living in their homes versus the alternative of nursing homes:


*“It’s a generation that has been hard for them to get what they have, and it seems to be hard for them get rid of all that. Despite being helpless, unprotected, and needing help from others”. *

*2NCMPOST*



*“You walk in [into the residence] and you see the little lounge full of people sitting in wheelchairs or armchairs… well, your world falls apart.” *

*NCM9*



*“The freedom you have in your home, you do not find it in an institution.” *

*NCM5POST*


In addition, they crave more respect and consideration for older people, experiencing helplessness and resignation for them—in their judgment—due to the detriment to the integration or presence of these people in family life:


*“I remember my grandmother when she fell ill […] she had to come to my house […] I had to take care of the grandmother”. *

*NCM5*



*“Today, what family is not working? […] The children have moved away from home; they cannot give them (elders) that support, that presence they (elders) need”. *

*2NCMPOST*


### 3.3. But… They Don’t Want (This) Care

Despite the empathetic understanding shown, resignation and attrition appeared when, in their daily wishes and effort to highlight risks and vulnerabilities and to offer guidance and potential solutions, NCM perceived that it was often a job that did not lead them to the desired outcomes:


*“But with the girl (caregiver) she has, she’s not comfortable either. Neither does she know what she wants, you say ‘Come on! let’s walk’ and she says ‘No, I can’t walk’. And the woman is more dependent than what she really is, and it is because she does not want to.” *

*NCM4*



*“I believe that it causes us wear and tear, we, the healthcare professionals, those who go periodically to the home and see the situation that they are living, you see that some have in their hands a possible solution.” *

*NCM8*



*“This woman […] it’s taken me years to try to get her into a residence because of the circumstances of her brother and her, and they can call you 24 times.” *

*1NCMPOST*


Even when they come to an apparent agreement because they receive resources (such as home help or a place in a residence), sometimes the EAR return, of their own accord, to the starting point:


*“And the caregivers, and you got there, and you convinced her, and the caregivers, and the appeal arrives, and the elder refused to accept (the resources). That wears out a lot.” *

*1NCMPOST*



*“‘What do you do, J, here?’ the elder says ‘I came here […] I can do nothing there [in the residence]’. I said: what the hell? after all that we have gone through!” *

*NCMPRE3*


Despite the disappointment and frustration, explained by the change in the attitude of the elder and the lost opportunity to improve the situation, the NCM were aware that they could not do anything about the necessary respect for the principle of autonomy of the person, hovering everyday between resignation and helplessness:


*“There are many rather regrettable situations because the decision of the patient’s right of autonomy prevails until he or she becomes disabled.” *

*NCM8*



*“They recommended that they should operate to get the possibility of an improvement [of the oncology process]... And the patient was not operated on for not leaving the cats alone.” *

*NCMPRE2*


### 3.4. The Help of the System Is (Outrageously) Limited

Resignation and helplessness, marked in the previous pattern, turned into indignation when the NCM affirmed that the resources available—mainly economic, those linked to the Law of Dependence and residences—were not enough, and that those available should be better distributed and supervised if the administration wanted to show an honest commitment to the elderly.

In this sense, they considered incomprehensibly unfair and shameful—even using the term *insane*—the fact that the care stipulated and approved under the Dependency Law took much longer than appropriate to implement:


*“To the serious case in which the dependency has to act, it is not immediate. A dependency law cannot take over a year, because there are cases that cannot wait over a year to resolve.” *

*NCM5POST*



*“The care based on the Dependency Law was requested and when it was approved, the patient had already died, after a year and a half.” *

*NCM5*


Indignation was accompanied by perplexity and shock because rapid responses to urgent problems had a response time of at least two months approximately (without exceptions), and by certain paradigmatic cases, such as that of the elderly person living alone, with blindness, being insulin-dependent and with a rudimentary cookstove; or other situations, such as those of homes with “mountains” of dirty clothes on the floor, or those in which the elder went upstairs crawling along and came downstairs on their backside, or in those who only ate muffins dunked in milk, or where they suffered abuse by their children, or where they ended up dying alone… circumstances, among others, that made the NCM privileged witnesses of how the social issues affect health (and vice versa).

### 3.5. Therefore, We Go beyond Our Limits

The set of affective experiences that led to indignation moved the participants to transcend limitations and go beyond what was stipulated by their role because of a compassionate, altruistic movement. Therefore, the NCM assisted in the kitchen, in cleaning the homes of the elderly and offered their telephones, accepting calls in their personal time:


*“It’s not just giving nutritional information; it’s telling them how to make croquettes [laughs]. Poor soul...”. *

*NCM8*



*“I have cleaned up many houses of Diogenes with C, the social worker […] and I say ‘C, let’s take away a little shit that here the cockroaches eat it and eat it all’ And I left carrying six bags of trash in each hand.” *

*NCMPRE3*



*“Sometimes, weekends come, I am usually available on my cell phone, and I have told them that I have it here if they need to call me.” *

*2NCMPOST*


However, this pushing the boundaries involved a cost, fatigue —a fatigue that they recognize equally in the team—and being aware of being at the limit of their strength, together with the necessity of accepting their own limitations, of not being able to be stretched far and wide:


*“I feel more tired, okay? And I think the team is much more tired too. We are stretched to the bone; we are fed up with not being seen and confronting many situations that, maybe at another less stressful time, we would have responded differently.” *

*NCM5POST*



*“Well, many times you go home saying, ‘You can’t do more,’ many times, I would have to take the person who is lonely to my home’.” *

*NCM9POST*


Moreover, the NCM considered that neither their duties nor going the extra mile in performing them were recognized by the administration nor by their fellow family nurses, which made them feel disappointed:


*“Disappointment sometimes yields despair, in addition to the lack of recognition […] I also feel it. […] starting with the administration that since 2002 has not been able to establish a position of nurse case manager, to that, to the colleagues and for the unawareness […] it gives me the impression that they think that our job can be performed by anyone”. *

*2NCMPOST*



*“Because I don’t see support from anywhere except the users and few colleagues. But regarding those who should be supporting me (my company, my immediate superiors, my nurses… for my part, zero [no help or recognition].” *

*NCMPOST5*



*“Even though our bosses don’t know what we do, the goals they set have nothing to do with us, despite everything they squeeze us, apart from everything they want to get out of us.” *

*NCMPOST2*


Hence, they expressed the need first for themselves and then for the whole community they served to become aware of and believe in their role and positive impact. However, they were also able to appreciate some recognition from the NCM themselves and from the EAR they cared for, committing themselves to the EAR despite everything described:


*“Have a conscience, that is, if I have those patients, first of all, for justice, we must care for them. Yes, the one that needs it most is the one that you have to support for justice, and then [if we don’t care for them], that will generate you more complications, more work, and a worse health situation of the population. For this reason, for conscience and for justice”. *

*NCM6*



*“We would need more uuummm no rewards, […] but at least recognition by the population, […] people should encourage us more, I think…”. *

*1NCMPOST*



*“I believe that we all believe in it, and here we are, we believe in it, and the elders believe in what we do, acknowledge it, and know we are here.” *

*NCMPOST1*



*“And, despite everything, we still set our shoulders to the wheel.” *

*NCM5POST*


## 4. Discussion

As the results of the study indicate, the NCM stand out in the attention to the EAR and give a relevant place to the existence of an interdisciplinary health-care team, where cooperation and a frequent, accessible, and horizontal communication should prevail [56].

This recognition of the work of the health-care team by the NCM is not strange. Previous studies [57,58] describe how the health-care team reduces the number of hospitalizations and readmissions and delays admission to residences. Furthermore, it relieves the burden of care on non-professional caregivers and improves patient satisfaction with the system.

In addition, the NCM in our study recognize the positive impact of the team on their own work and satisfaction. This result is consistent with previous studies reporting how the satisfaction of NCM increased when their professional experience was recognized by other disciplines [2,59].

However, beyond the health-care team, NCM highlight the relevance of context to the care of EAR. In this sense, the study conducted by Macias-Colorado et al. [60] highlights how case managers considered factors related to the patient’s environment and primary caregiver to be crucial to the patient’s safety, highlighting three of these factors: (1) the need for the caregiver to accept and assume her role, (2) recognition of her work by other family members and, consequently, to help her in her rest and self-care, and (3) an adequate economic situation. These authors do not mention the problems of accessibility and barriers in housing, as different studies [61,62] have reported and as described in the results of our study.

An important aspect that appears in the context is the gender role. The NCM recognize the role of women in EAR care. Despite the naturalization/normalization of this role, this recognition is visible in our study. As it is evident from the literature, NCM are fully aware that women continue to follow, for the most part, the dictates of the femininity that underpin empathy and care for others, prioritizing the well-being of others, dealing with the housework and the care of the elderly and the sick, so they lack time for themselves and for activities that benefit their health [63]. Meanwhile, men have increased their participation in parenting activities in recent years but are hardly involved in caring for older and dependent persons [64,65]. In the same sense, another nuance reflected in the discourse of the informants, and in the literature, is that NCM identify a higher vulnerability when the caregiver is an older man [66].

Another of the allies of EAR care and informal caregivers is the neighborhood, which is the last recognition the NCM make in the present study. When it exists, it becomes an essential resource supporting many (informal but also professional) caregivers in the care they perform, especially in the so-called emotional accompaniment [67,68].

In addition, the NCM participating in our study give special relevance to the context when they identify the absence of family care or neighborhood support [62,68,69], which is an increasingly frequent situation, as indicated by the literature, in places such as Spain, where the atomized neoliberal society model has been replacing the Mediterranean model of care in which the family plays an essential role [70,71,72].

Nevertheless, and despite these difficult circumstances of the context, 87.3% of older people prefer to live in their home, even if they are alone [62,69], to the alternative of living in a residence or with other relatives [62].

According to the results of the study, this decision is shared by the NCM. In addition, this preference of the elderly to live in their usual environment, in their own home, is not only recognized by NCM but is a recommendation (also recognized by the WHO [73]).

However, this understanding is not always easy. The NCM feel resignation and attrition when the EAR refuses care, such as home caregivers or a place in a residence. Campbell et al. reported [74] how older adults often choose not to accept recommended social support services.

The study results describe how case managers and other professionals, such as social workers, are often caught up in the conflict of encouraging patients to accept services they do not want. At the same time, they accept that such refusal means maintaining situations of insecurity in the home, for example, after discharge from the hospital. According to Reinhoudt-den Boer, van Wijngaarden and Huijsman [75], these conflicts are due, in part, to the mistaken assumption that the various parties involved share a set of beliefs and values about the care and interventions offered. However, as they refer in their speeches, the NCM are resigned to these refusals, basically, because of the recognition of the principle of autonomy. These experiences are reflected in the study of Campbell et al. [74] who state that case managers have the task of organizing support services for older adults, balancing their right to self-determination/autonomy with the maintenance of their security and that of the community. 

Reinhoudt-den Boer et al. reflected [75] the emotional experience of these professionals when they find how their negotiation attempts do not led to the expected results. Recent studies [74,76] describe interventions related to the humanization of care and negotiation that are capable of overcoming these conflicts.

Regarding the fourth pattern (and the help of the system is (outrageously) limited), indignation, shame, perplexity, and shock are easy to understand when their experiences are endorsed by different publications. Spain has traditionally been one of the countries of the European Union that have allocated less public funds to family support [72]. Despite the implementation of the Dependency Law in 2006 [23], the resources that the state makes available to the elderly are still far from sufficient. However, this order of things is also echoed at the international level. According to Teper et al. [3], resource constraints and ineffective health administrations—with access to many of these resources restricted to physician approval, budget constraints, or unrealistic guidelines—make care plans and interventions by professionals such as NCM difficult. Furthermore, today’s European health systems tend to offer fragmented care to older people. Many of them receive this care from numerous specialists or through different levels of care [3].

Thus, in this context, where public policies for the care of dependent elders are weak, the mandatory altruism practiced by women and the solidarity of citizens mentioned here in the second pattern becomes more necessary [65]. However, this compassionate, altruistic movement reaches out to the one conducted by the NCM of the present study when they go beyond what is stipulated in their role. In this sense, different studies [4,77] also describe this going beyond the limits in NCM to the care of other population profiles, pointing to a necessary balance between multiple roles (especially those of providing direct care, being a liaison and training others), while, because of overload and this multiplicity of roles, they were forced to extend their working hours. This way of feeling, doing, and going beyond that is shown by nurses and NCM has gained special relevance during COVID-19, as is reflected in the literature [57].

However, this going beyond the limits, which appears in the discourse of the informants of this study, involves a cost. As previous studies reported, NCM experience fatigue, frustration, and disappointment motivated by a lack of understanding of the health system/organization and their own colleagues. Although most scientific literature in this area is focused on hospital services, it is known that the care of NCM in complex settings such as those described may have a negative impact on their mental and emotional health, among other factors, because of the discrepancy between the efforts made and the achievements, which can lead them to suffer burnout or compassion fatigue [37,78].

Among the factors contributing to this situation, Teper et al. [3] highlight some that are also found in our study, such as: time pressure, excessive bureaucracy, impersonal methods of communication, unexplained or sudden changes in protocols, distrust and medical resistance to the figure of NCM, or when other disciplines (even nursing) believe that case management is redundant or doubt that it will improve patient outcomes.

In short, the maintenance of the emotional labor derived from the context where the NCM conduct their work involves an emotional cost affecting their behavior and their care (reaching even, as other authors reported, dysmenorrhea, sleep disorders, and suicidal thoughts [79]), preferring, for example, avoidance to the affective connection with patients [66]. At this point, the gender role appears again: Considering that the great majority of NCM are women, it is advisable to observe that they are the ones that, according to the established by femininity, seem to be more emotionally involved than men and, therefore, they are more prone to perform a larger emotional labor [79].

Finally, we highlight the positive message derived from the speech of the NCM who, despite all the difficulties, are betting on continuing to set their shoulders to the wheel. This feeling of commitment has been studied in nurses whose sense of duty has led them to remain involved in day-to-day professional care, being considered simply a duty merely because they are part of the nursing staff [76]. Another incentive is that when they can humanize the patient, they increase the emotional and cognitive connection, providing satisfaction in their performance [76]. One of the biggest incentives for commitment is to be recognized for the results (in all areas of their performance [1,77,80,81]) and for the care itself by their colleagues (whether or not they are case managers), patients, and relatives [59,82].

### Strengths and Limitations of the Study

In our opinion, the present study has been able to make visible the emotional experience of the NCM in the care of EAR in Spain with a comprehensive approach, which has placed the focus on the (traditionally considered) negative emotions and on the positive ones. We believe that the approach to the understanding of the emotional experience contributes to the claim of their role and to face the limitations and pressures of the current social and health environment.

However, this comprehensive understanding of the affective phenomenon has not become complete. Emotions such as satisfaction, pleasure for achievement, admiration, confusion, and hope have not yet saturated the discourse analyzed but were present in interviews and in the early levels of analysis. Therefore, understanding the phenomenon would benefit from a new approach more focused on these emotions in particular.

In terms of methodological considerations, the fact that the lead author worked as an NCM in the same district as the participants forced her to undergo a process of awareness of her role throughout the investigation process, especially during data collection and analysis, which will avoid a hasty understanding conditioned by her own subjective projection.

Regarding collecting information, it should be noted that most interviews were conducted during working hours and individual interviews were conducted in the health-care centers. These circumstances may have influenced the care and time availability of NCM.

Finally, and regarding the analysis, it should be noted that an interpretative approach has been maintained, even though finding signs pointing to the opportunity to open sociocritical approaches that would allow suggesting changes. However, this option would have opened up expectations that would have been difficult to manage without the involvement of other agents who are not currently included in the study.

## 5. Conclusions

The results of the present study bring us closer to the following synthesis of the emotional universe of NCM for the care they provide to the EAR in PHC: firstly, these professionals recognize the social phenomenon, the social–health team, the women caregivers, and a close and supportive environment as natural allies in their role; secondly, the NCM “connect” with the desire of the EAR to stay at home through compassion; thirdly, they state that patients often do not listen to their suggestions, which causes them frustration and helplessness; then, these emotions are transformed into perplexity and indignation when they perceive the significant limitations of the health system; as a consequence, and finally, the NCM stated that they went beyond the limits of their role to try to fill gaps in a display of pride.

One of the ideas implicit in the study results is that the events experienced during the pandemic have only underscored the situation that had been proven before its appearance. This situation was marked by: (1) the difficult life situation of the elderly at risk and (2) the even more pressing limitations and pressures on the NCM in the current scenario of the precariousness of the social and healthcare context.

The complexity of the situation requires further research on the emotional and social determinants of the phenomena. The resulting evidence will compel clinicians, organizations, and governmental institutions to recognize the work of professionals with this population.

Furthermore, it would make visible the need for more interdisciplinarity, counseling, and inter-institutional coordination. These actions would speed up responses to this context of vulnerability and avoid referrals or referrals to the public prosecutor’s office. However, the real solution to this situation is the involvement of everyone, as a society, in improving the attention to the EAR.

Finally, it would be pertinent to explore the particular interaction between years of experience as an NCM, gender, and emotions, which are issues that the present study has failed to satisfy.

## Figures and Tables

**Figure 1 ijerph-19-16445-f001:**
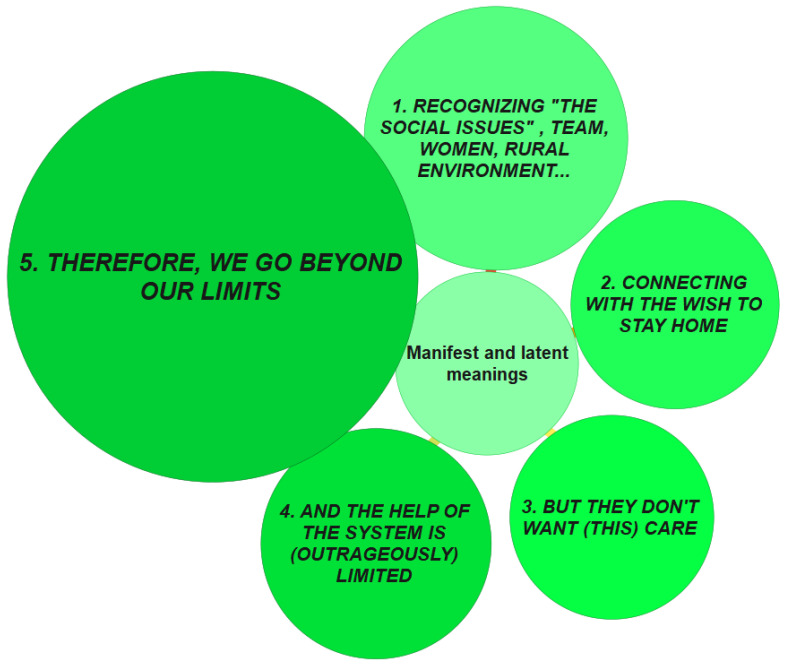
Explanatory framework of results.

**Figure 2 ijerph-19-16445-f002:**
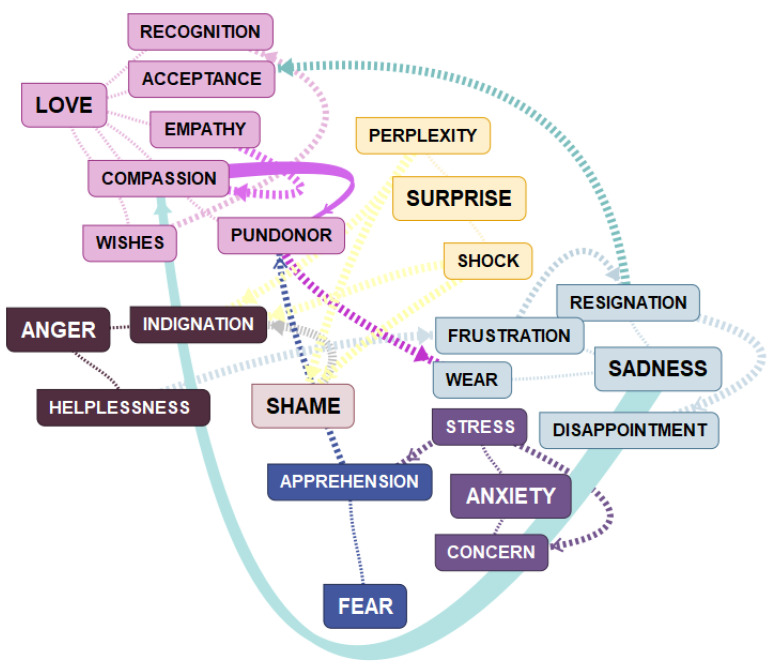
Emotional Universe of NCM in the care of elderly people at risk in Spain.

**Table 1 ijerph-19-16445-t001:** Script of the semi-structured interview and focus group interview. NCM: nurse case managers.

Questions related mainly to the emotion of Recognition
Tell me, in this post-pandemic period, how do you see yourself—as NCM—and how do you see the team?
Hey, and social affairs, how do you handle them? They seem to be the most notorious thing in our care. I don’t know what you think about it…
And about the environment, what can you tell me? What is its role? (rural/neighborhood)
Tell me about the role of elder women versus elder men
**Questions related mainly to emotions of Empathy, Compassion, Wish, and Attrition**
Do you know that, despite being complicated, it seems that people would really prefer to be at home? What do you think about this?
We are there, facing a feeling of loneliness and tremendous neglection... How do you see it?
What would you ask for about it?
Sometimes we don’t know how to solve the problem, right? Neither the patients, nor the family, nor the professionals... Sometimes, they don’t want help, and we wear down... I don’t know if you can comment on it...
**Questions related mainly to feelings of Frustration and Helplessness**
How do you manage the feeling that you can sometimes do nothing (e.g., they don’t want what is offered to them, that social issues are the main problem, and that the health-care center has its limitations...)?
**Questions related mainly to the emotions of Perplexity, Shock, and Outrage**
Do you know that it seems that there are still personal and family situations that do not come out in the newspapers but that we know well?... Situations that make an impact... I don’t know if I’m wrong
And as for dependency care, some colleagues describe it as insane to take so long to come... how do you see it?
**Questions related mainly to the emotions of Self-recognition (appreciation) and Acceptance**
Other NCM say we need to be aware (appreciate) and believe in what we do. How are we doing in this regard?
Although we cannot reach everything, can we?
**Questions related mainly to the emotions of Pride, Admiration, and Hope**
What makes it that, despite everything, it can still be worth betting on the elderly?
Who do you admire, and why, when it comes to this fragile situation?
Is there hope? What is your hope like?

**Table 2 ijerph-19-16445-t002:** Criteria of rigor and quality. NCM: nurse case managers.

Criterion	Description
Theoretical, epistemological, and methodological adaptation	The qualitative paradigm allows for an open, flexible, and changing design. Within it, interpretative phenomenology is most suitable if it is intended to recognize the primary relevance of emotional experiences in phenomena related to care professionals. This perspective highlights the (also emotional) involvement of the research team in what they are observing and in the other dimensions of the world of life to which affections can act as a gateway. A hermeneutical analysis facilitates the transferability of the study to contexts other than the one studied.
Relevance	The NCM face a very significant diversity of emotions during their professional practice. These emotions arise from the crucial events they experience daily in patient and family care and the coordination/management of care. The study of this emotional world is relevant for various reasons. First, there is a concern for the health and well-being of these professionals as a consequence of their work. Secondly, there are the essential individual and organizational phenomena that several authors describe as a consequence of emotions at work—or of the more recently called emotional labor (as an intrinsic characteristic of the job). Eventually, these emotions have been shown to play a fundamental role in aspects such as job satisfaction, group functioning, or leadership.
Validity	Through a triple triangulation of the study; results were found: (1) between the lead author and participants, (2) between the research team, and (3) between different exploratory techniques: individual interview and group interview. In this sense, the resulting transcripts, in verbatim, were returned to the subjects for validation and comment. During the analysis, the author was assisted by a co-author who made a parallel analysis of the first interview. In addition, after completing each level, both agreed on their decisions and shared them with the remaining co-authors. Finally, after the third level, they also shared their decisions with the participants. Nevertheless, the hermeneutical arch of the third level of analysis would be completed, at first, with the elaboration of the definitive explanatory framework of results (also agreed) and, secondly, with the discussion of the study findings. To ensure transferability (or degree of applicability of the results to other contexts, situations, and participants), the authors provided a detailed description of the study setting.
Reflexivity	All authors are nurses and feel particularly motivated and sensitive to the psychosocial aspects of nursing care and research—specifically, the relationships between work environment and emotional health. Furthermore, the lead author has been working for 19 years as an NCM in the health-care setting of the study, working directly with NCM as the interviewees (whom she knows personally). This emic perspective involves, a priori, sharing emotions linked to their performance. To ensure the richness of meanings generated by intersubjectivity and avoid biases, the author performed an affective self-hermeneutical process that she would periodically reflect on her research notebook and share with the rest of the co-authors.

**Table 3 ijerph-19-16445-t003:** Socio-demographic profile of the participating NCM.

PRE-PANDEMIC PERIOD		Gender	Age	Experience (in Years) as an NCM	District
Individual interview participants					
	NCM5	F	54	17	Guadalquivir
	NCM6	M	51	17	Guadalquivir
	NCM9	F	55	4	Córdoba
	NCM4	F	51	14	Guadalquivir
	NCM8	F	48	7	Córdoba
Discussion group participants					
	NCMPRE1 *	F	48	10	Córdoba
	NCMPRE2	F	61	15	Córdoba
	NCMPRE3 *	F	53	17	Córdoba
	NCMPRE4 *	F	52	18	Córdoba
	NCMPRE5 *	F	54	8	Córdoba
	NCMPRE6 *	M	60	18	Córdoba
**PANDEMIC PERIOD**		**Gender**	**Age**	**Experience (in years) as an NCM**	**District**
Individual interview participants					
	NCM9POST	F	57	6	Córdoba
	1NCMPOST	F	51	15	Córdoba
	NCM6POST	M	52	19	Guadalquivir
	2NCMPOST	F	54	20	Córdoba
	NCM5POST	F	55	19	Guadalquivir
Focus group participants					
	NCMPOST1	M	57	8	Guadalquivir
	NCMPOST2	F	58	19	Córdoba
	NCMPOST3	F	55	19	Córdoba
	NCMPOST4	F	57	20	Córdoba
	NCMPOST5	F	56	6	Córdoba
	NCMPOST6	M	62	20	Córdoba

* Similar identities, which are not explicit by encoding, are as follows: NCMPRE1 and 1NCMPOST; NCMPRE4 and 2NCMPOST; NCMPRE3 and NCMPOST3; NCMPRE5 and NCMPOST5; NCMPRE6 and NCMPOST6. NCM: nurse case manager; F: female; M: male. PRE: pre-pandemic period. POST: pandemic period.

## Data Availability

Not applicable.

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
