# Peer review of "Emotional Universe of Nurse Case Managers Regarding Care for Elderly at Risk in Spain: A Hermeneutical Study"

_ijerph, 2022, doi:10.3390/ijerph192416445_

Round 1

Reviewer 1 Report

Title

Need to add the study design information in the title. It will make the reader easier to identify the study.

Abstracts

1.     The aim & the study result of the study in the abstract sections was not clear. Please kindly reorganize the aim & study result of the study to be clearer in the abstract section. The main study result must be clearly stated in the abstract section.

Introduction

2.     Paragraph 1

-       The sentence on page 1 lines 29-32 is too long.

-       There are no references for page 1 lines 30-34

Paragraph 3 (pages 1-2 lines 40-49)

-       What about life expectancy, chronic diseases & their consequences in Spain?

-       What makes Spain different from other countries?

-       The sentence on pages 1-2, lines 43-49 is too long

Paragraph 4 (page 2 lines 50-54)

The authors mentioned the role of NCM in the issue of life expectancy, how about in Spain?

Paragraph 5 (page 2 lines 55-64)

The authors mentioned the role of nurses in NCM (PHC) including in Spain,

but do they have any obstacles to playing the role? These questions may be related to the description in paragraph 7.

Paragraph 7 (page 2 lines 74-81) effective experience of NCM

-       The authors have mentioned that NCMs are often overwhelmed, but does this issue only impact their emotions? How about their health physics? Why the emotional issue be the most important issue to be raised in this study? Please add literature to highlight the urgency of emotional issues among NCMs in EAR care.

-       The sentence on page 2 lines 74-78 is too long & no reference is provided.

Overall: scientific reasons, the gaps, and the novelty in the introduction sections need to be more stressed in the introduction section. The author also mentioned in the method section that the study has been conducted in the Andalusia province, why did the author choose this location?

3.     Method

-       How many sample sizes were employed in this study?

-       Individual and group interviews, how did the author conclude the result from this different interview method?

4.     Result

-       Since the characteristics of the participants vary including gender distribution and NCM’s year of experience, how does the author control such potential confounder variable? Moreover, the author found that gender was also the main issue in the findings.

-       The data collection of the study included the period before and after covid 19. How do the authors conclude the findings since the situations are different?

5.     Discussion

-       Please provide the summary of the key results with reference to study objectives

-       Since the study has been conducted in Andalusia province, could the study findings be generalized to all NCMs across Spain?

-       The sentence on page 14 lines 435-438 is too long. Please reorganize it.

-       The sentence on page 15 lines 483-487 is too long. Please reorganize it.

6.     Conclusion

-       The conclusion is too long.

-       What is the impact of the study on future research, clinicians, or related government who establish the policies for the NCM?

Author Response

In response to reviewer 1's comments, the authors would like to comment on the following:

First, the authors would like to thank the reviewers for their interest in our work and for the time and attention devoted to the review process, as we believe the article has been substantially improved. Next, the authors highlight the reviewers' comments in bold characters and then offer their responses.

Title

Need to add the study design information in the title. It will make the reader easier to identify the study.

Thanks for the suggestion. The authors have added the following to the title (followed by a colon): "a hermeneutical study".

Abstracts

The aim & the study result of the study in the abstract sections was not clear. Please kindly reorganize the aim & study result of the study to be clearer in the abstract section. The main study result must be clearly stated in the abstract section.

Thanks for the indication regarding the summary of the study. Due to editorial regulations, the authors have not explicitly included sections within the study, although they have structured it taking them into account. However, the authors have added a sentence emphasizing the coherence between the objective and results. All in all, it is shown in the manuscript as follows: “Therefore, it is necessary to understand the Emotional Universe (EU) of NCM regarding the care they provide to EARs in Primary Health Care, as well as the reasons that generate these emotions”.

Introduction

Paragraph 1

  •       The sentence on page 1 lines 29-32 is too long.

Thanks for the observation. Undoubtedly, it will contribute to a more precise and smoother introduction. The authors have considered splitting the sentence in two as follows:

The role and collaborative process of nurse case managers (NCM) consist of the following five constitutive actions to meet health needs, as described in the scientific literature: to assess, plan, implement, coordinate, and value options and services [1]. This role involves rarely visible emotional phenomena, such as emotional labor, well-being, satisfaction, and inspiration that can derive from it. 

-       There are no references for page 1 lines 30-34

Thanks for the observation. The authors have added the references found where there is evidence of the emotional phenomena associated with the work of these nurses (see manuscript). However, the documentary's silence on the subject is such that not even this invisibility is included in the references, which, we believe, adds originality and relevance to the present study.

Paragraph 3 (pages 1-2 lines 40-49)

  •     What about life expectancy, chronic diseases & their consequences in Spain? What makes Spain different from other countries?

Thanks for the issues raised. The authors have contextualized the phenomena in Spain adding the next highlighted text:

In the Spanish context, in a few decades, this country has positioned itself in the world's leading group in life expectancy at birth, and by 2040 it could lead it, ousting Japan from first place. However, this increase in life expectancy leads to an increase in the number of chronic diseases or accidents occurring in the final years of life (in line with the rest of the industrialized countries). Resulting, sooner or later, in some type of disability that, in turn, will diminish personal autonomy, causing an increase in the demand for care, especially in long-term care. In Spain, like the rest of the leading European countries, most chronic diseases fall on the elderly. Furthermore, the most significant healthcare expenditure incurred by individuals is concentrated in the last years of their lives; the older population naturally (or automatically) becomes the greatest beneficiary of public healthcare.

-       The sentence on pages 1-2, lines 43-49 is too long

Thanks for the indication. The authors have split the sentence as follows:

Currently, the Primary Health Care Systems (PHC) provide this care mainly in the home of the elderly. The PHC prioritizes the so-called EAR patients. These are identified as dependent elderly—or at risk of dependence—older than 80. EAR patients may present some or several of the following characteristics: living alone, isolated or without a competent socio-family network, with housing and economic problems, possible victims of abuse, or at risk of premature or inadequate institutionalization.

Paragraph 4 (page 2 lines 50-54). The authors mentioned the role of NCM in the issue of life expectancy, how about in Spain?

Thanks for the question raised. The authors added the following text (after the reference to the global context): "as it is in the Spanish context". This addition was accompanied by appropriate bibliography.

Paragraph 5 (page 2 lines 55-64). The authors mentioned the role of nurses in NCM (PHC) including in Spain, but do they have any obstacles to playing the role? These questions may be related to the description in paragraph 7.

Thanks for the observation. The authors have pinpointed the situations (precisely) identified in the article's results —which generate their emotional response— as the obstacles for these professionals to play their role.

Paragraph 7 (page 2 lines 74-81) affective experience of NCM. The authors have mentioned that NCMs are often overwhelmed, but does this issue only impact their emotions? How about their health physics? Why the emotional issue be the most important issue to be raised in this study? Please add literature to highlight the urgency of emotional issues among NCMs in EAR care.

Thanks for the questions and suggestions raised. Undoubtedly, the reasons that make the present study necessary and pertinent are, with them, more explicit. The authors have included the following text in response to them:

The most visible work of nurses is the provision of social and health care in situations of high vulnerability, which involves emotional labor that is rarely recognized, even by the professionals who suffer it. However, the consequences are not only worrying in the psycho-affective field —with high emotional suffering, compassion fatigue, and burnout— but also in the physiological field —with cardiovascular manifestations, such as arterial hypertension or musculoskeletal type problems— and even in the professional development of nurses, even undermining their work competence. Moreover, this aspect is closely linked to emotions because the Emotional Determinants of Health are considered fundamental due to their influence on aspects such as job satisfaction, teamwork, leadership, or decision-making processes, among others.

These consequences, which have been widely described in the scientific literature when referring to nurses in general, have not been addressed concerning their affective experience or its determinants, especially in NCM.

  •       The sentence on page 2 lines 74-78 is too long & no reference is provided.

Thanks. The authors have answered this suggestion in the previous question.

Overall: scientific reasons, the gaps, and the novelty in the introduction sections need to be more stressed in the introduction section. The author also mentioned in the method section that the study has been conducted in the Andalusia province, why did the author choose this location?

Thanks for the summary of comments associated with the introduction. The authors have previously responded to the above. Regarding the choice of location, the authors describe the reasons for this in Table 2 (Rigor and quality criteria), specifically in the "Reflexivity" section.

  1. Method
  •       How many sample sizes were employed in this study?

Thanks for the observation. The authors understand that this aspect may be confusing due to the two different moments of the study. However, their initial sample size is that of the NCMs identified in the data corpus of the thesis from which it starts, i.e., 11 NCMs. Vital and contractual changes suffered by three of the participants in the pre-pandemic period (NCM4, NCM, and NCM PRE2) meant that they were unable to participate in the second moment, incorporating three new participants. Hence, 14 NCMs participated in the two periods as a whole.

  •       Individual and group interviews, how did the author conclude the result from this different interview method?

Thanks for the reference to the different data collection tools because of the implications they may have on the results. The primary tool of the study was the individual interview. At both moments, the group interview was used as a method of triangulation. In the authors' criteria, the differences between the two types of discourse (individual and group) did not reach saturation.

  1. Result
  •       Since the characteristics of the participants vary including gender distribution and NCM’s year of experience, how does the author control such potential confounder variable? Moreover, the author found that gender was also the main issue in the findings.

Thanks for highlighting two characteristics (experience and gender) that may be relevant when interpreting the participants' different experiences. However, the analysis of the participants' discourse could not show differences. Nevertheless, thanks to this observation, the authors can add suggestions for future lines of research specific to this purpose (see the Conclusions section).

  •       The data collection of the study included the period before and after covid 19. How do the authors conclude the findings since the situations are different?

Thanks for making explicit an interest shared from the outset by the authors. One of the ideas that underlie the reading the study results is that the events experienced in the pandemic period have only underscored the situation that was proven before its appearance: the challenging life situation of the elderly at risk, on the one hand, and the even more pressing limitations and pressures suffered by the NCM in the current scenario of the precariousness of the socio-health context, on the other.

  1. Discussion
  •       Please provide the summary of the key results with reference to study objectives

Thanks for the indication. Despite its timeliness, we believe this summary already appears in two sections of the study: the summary results and the beginning of the conclusions. However, if necessary, the authors can include a paraphrase of this summary at the beginning of the discussion.

  •       Since the study has been conducted in Andalusia province, could the study findings be generalized to all NCMs across Spain?

Thanks for raising this question since, through it, the authors have realized that they did not mention the criterion of transferability of the results (even though they considered it throughout the process). In Table 2 (Rigor and Quality Criteria), in its Validity section, the following text will appear in this regard:

To ensure transferability (or degree of applicability of the results to other contexts, situations, and participants), the authors provided a detailed description of the study setting.

  •       The sentence on page 14 lines 435-438 is too long. Please reorganize it.

Thanks for the observation. Following it, the authors have considered structuring its contents as follows:

This recognition of the work of the healthcare team by the NCM is not strange. Previous studies describe how the healthcare team reduces the number of hospitalizations and readmissions and delays admission to residences. Furthermore, it relieves the burden of care on non-professional caregivers and improves patient satisfaction with the system.

-       The sentence on page 15 lines 483-487 is too long. Please reorganize it.

Thanks for the observation. Following it, the authors have considered structuring its contents as follows:

The study results describe how case managers and other professionals, such as social workers, are often caught up in the conflict of encouraging patients to accept services they do not want. At the same time, they accept that such refusal means maintaining situations of insecurity in the home, for example, after discharge from the hospital.

  1. Conclusion
  •       The conclusion is too long.

Thanks for the observation. Following it, the authors have considered reducing its content as follows (its first paragraph appears below while what follows appears as an answer to the next question):

The results of the present study bring us closer to the following synthesis of the emotional universe of NCM for the care they provide to the EAR in PHC: firstly, these professionals recognize the social phenomenon, the social-health team, the women caregivers, and a close and supportive environment as natural allies in their role; secondly, the NCM "connect" with the desire of the EAR to stay at home through compassion; thirdly, they state that patients often do not listen to their suggestions, which causes them frustration and helplessness; then, these emotions are transformed into perplexity and indignation when they perceive the significant limitations of the health system; as a consequence, and finally, the NCM stated that they went beyond the limits of their role to try to fill gaps in a display of pundonor.

  •       What is the impact of the study on future research, clinicians, or related government who establish the policies for the NCM?

Thanks for the indication. The authors have prefaced their response with the reflection on the impact of the pandemic made above. The following closure will be included in the manuscript:

One of the ideas implicit in the study results is that the events experienced during the pandemic have only underscored the situation that had been proven before its appearance. This situation was marked by: 1) the difficult life situation of the elderly at risk and 2) the even more pressing limitations and pressures on the NCM in the current scenario of the precariousness of the social and healthcare context.

The complexity of the situation requires further research on the emotional and social determinants of the phenomena. The resulting evidence will compel clinicians, organizations, and governmental institutions to recognize the work of professionals with this population.

Furthermore, it would make visible the need for more interdisciplinarity, counseling, and inter-institutional coordination. These actions would speed up responses to this context of vulnerability and avoid referrals or referrals to the public prosecutor's office. However, the real solution to this situation is the involvement of everyone, as a society, in improving the attention to the EAR.

Finally, it would be pertinent to explore the particular interaction between years of experience as an NCM, gender, and emotions, issues that the present study has failed to satisfy.

Reviewer 2 Report

Overall, this is a very interesting paper.

On page 2, line 55, it states "Nurses are the professionals who mainly perform this work in PHC" It is not clear what this work is.

The questions in the interview guide appear quite leading. This may simply be a translation issue, and perhaps the original questions were not as leading? As they are currently written, they show bias. If these are the actual questions that are used, this bias should be addressed in the paper.

The framework pictured in Figure 1 does not help illustrate the results. Again, this may be due to translation issues. Suggest removing the framework or making it clear.

Author Response

In response to reviewer 2's comments, the authors would like to comment on the following:

First, the authors would like to thank the reviewers for their interest in our work and for the time and attention devoted to the review process, as we believe the article has been substantially improved. Next, the authors highlight the reviewers' comments in bold characters and then offer their responses.

Overall, this is a very interesting paper.

On page 2, line 55, it states "Nurses are the professionals who mainly perform this work in PHC" It is not clear what this work is.

Thanks for the observation. It refers to the role of the Care Manager. The authors have changed it in the manuscript to clarify its meaning.

The questions in the interview guide appear quite leading. This may simply be a translation issue, and perhaps the original questions were not as leading? As they are currently written, they show bias. If these are the actual questions that are used, this bias should be addressed in the paper.

Thanks for the observation. The explanation of why the script has this formulation of its questions is related to the fact that it was designed from the results coming from a previous analysis of the pre-pandemic data corpus. Then, the purpose was to focus on further exploring the particular emotions and issues derived from this former analysis, though focus group interviews. In the manuscript, this process is described in these lines:

"An interview script was drafted for both individual and group interviews. In the case of the pre-pandemic script, the starting categories based on the taxonomy of the Universe of Emotions were used. In contrast, for the time of endemism, an interview script was drafted based on the results of a first analysis of the documentary data corpus consisting of the verbatim transcripts of the semi-structured interviews and discussion group conducted to the NCM corresponding to the thesis project (Pre-pandemic period) (Table 1)".

The framework pictured in Figure 1 does not help illustrate the results. Again, this may be due to translation issues. Suggest removing the framework or making it clear.

Thanks for the suggestions. Despite its opposite intention, the authors have realized that the figure can confuse readers due to its volume of contents: main themes plus related main categories. The authors have decided to simplify it, depicting (only) the results main themes with various visual elements coherent with meanings associated with the study context and uncovered by the results analysis: the reasons that generate NCMs’ EU constitute the main themes of the present study.  These five themes are depicted over visual elements (circular and progressive green patterns) that relate with the following insight: the experiences lived during COVID-19 have intensified the context of the crisis before its appearance. First, the problematic life scenario of the elderly at risk, and then the even more pronounced limitations and pressures to which NCM are subjected in the current circumstances of the precariousness of the social and healthcare context (Figure 1). The text from “the reasons that generate…” to “… healthcare context” has been included in the manuscript. The main categories linked with the main themes are shown in the narration of the results.

Reviewer 3 Report

This paper is a qualitative study that identifies the emotional universe of nurse case managers for elderly persons at risk.

The authors described, with new findings, the emotions of nurse case managers, which are difficult to obtain in quantitative studies.

However, as a reviewer, I believe the paper would be even better if the following five points were revised.

1. In the abstract section, the importance of understanding the emotional universe of nurse case managers is described. However, what is known and what is not known from the existing research is not sufficiently described.

2. In the introduction section, I think the description is redundant and too long. The description prior to the following sentence is particularly, redundant and too long. "Therefore, the emergence of the role of NCM is considered paradigmatic, in this sense, at the global level."

3. In Table 1, NCM is not indicated what it is an abbreviation for. It should be described so that the reader can understand it by looking at Table 1 only. The following tables and figures should be described similarly.

4. The author should explain the figures so that the reader can understand them by looking at Figures 1 and 2 alone.

5. You should include any additional research you consider necessary in the future at the end of the conclusion section.

Author Response

In response to reviewer 3's comments, the authors would like to comment on the following:

First, the authors would like to thank the reviewers for their interest in our work and for the time and attention devoted to the review process, as we believe the article has been substantially improved. Next, the authors highlight the reviewers' comments in bold characters and then offer their responses.

This paper is a qualitative study that identifies the emotional universe of nurse case managers for elderly persons at risk.

The authors described, with new findings, the emotions of nurse case managers, which are difficult to obtain in quantitative studies.

However, as a reviewer, I believe the paper would be even better if the following five points were revised.

  • In the abstract section, the importance of understanding the emotional universe of nurse case managers is described. However, what is known and what is not known from the existing research is not sufficiently described.

Thanks for the observation. The antecedents of the phenomena and the contributions of the study are disseminated in the manuscript. Although it is not possible to attach these sections to the document due to regulations and formatting (e.g., word count), the synthesis of what is known and what is not known is shown here:

 What is known:

1) The elderly population at social risk is increasing; 2) This population group is a frequent user of NCM services; 3) The care provided by caregivers to this group generates a high emotional impact.

What is not known (Main contributions):

1) The results are useful for managers of health and social systems, as they show the existing problems in the care of this group; 2) The results are useful for NCM, as they identify not only emotions but also ways of coping with NCM; ); 3) The results are useful for society as a whole, as they show the need for a transversal collaboration all its members.

  • In the introduction section, I think the description is redundant and too long. The description prior to the following sentence is particularly, redundant and too long. "Therefore, the emergence of the role of NCM is considered paradigmatic, in this sense, at the global level."

Thanks for the suggestions. The authors have made amendments to the manuscript to reduce redundancy and add clarity. Accordingly, the authors correct the following text earlier in the introduction:

The role and collaborative process of the nurse case managers (NCM) consist of the following five constitutive actions to meet health needs, as described in the scientific literature: to assess, plan, implement, coordinate and value options and services [1]. This role involves rarely visible emotional phenomena, such as emotional labor, well-being, satisfaction, and inspiration that can derive from it.

The authors have also avoided redundancy and simplified the paragraph of the mentioned sentence:

Therefore, effective and efficient responses from various public administrations are required. Among these responses, there is the establishment of new nursing roles, with a significant deepening and expansion of their competencies. Therefore, the emergence of the role of NCM is globally considered paradigmatic in this sense.

  • In Table 1, NCM is not indicated what it is an abbreviation for. It should be described so that the reader can understand it by looking at Table 1 only. The following tables and figures should be described similarly.

Thanks for the observation. The authors have corrected this mistake.

  • The author should explain the figures so that the reader can understand them by looking at Figures 1 and 2 alone.

Thanks for the suggestions. The authors have added an explanation for each figure. Figure one has been simplified, as well, to gain clarity.

For figure one:

Reasons generating the EU of NCM. These five themes are represented by visual elements (circular and progressive green patterns) that relate to the following insight: the experiences lived during COVID-19 have intensified the crisis context existing before its occurrence. First, the problematic life scenario of the elderly at risk, and then the even more pronounced constraints and pressures to which NCMs are subjected in the current circumstances of the precariousness of the social and health context (Figure 1)

For figure two:

[These patterns describe different trajectories (according to their categories and sub-categories) within the emotional galaxies of Bisquerra (Figure 2)] . As the figure shows, in the EU of NCM, several emotions are directly linked, such as sadness-compassion, compassion-pundonor, and perplexity-shame-indignation. These associations are contextualized in the narrative description of the results below. On the other hand, absences stand out in the figure, such as the emotions belonging to the galaxies of happiness, joy, and disgust, which, although present in the participants' discourse, did not reach saturation.

  • You should include any additional research you consider necessary in the future at the end of the conclusion section.

Thanks for the indication. The authors have reworded the last paragraph of the conclusions to read as follows:

One of the ideas implicit in the study results is that the events experienced during the pandemic have only underscored the situation that had been proven before its appearance. This situation was marked by: 1) the difficult life situation of the EAR and 2) the even more pressing limitations and pressures on the NCM in the current scenario of the precariousness of the social and healthcare context.

In this sense, to the lines of research suggested in the limitations section, the following have been added:

The complexity of the situation requires further research on the emotional and social determinants of the phenomena. The resulting evidence will compel clinicians, organizations, and governmental institutions to recognize the work of professionals with this population.

Furthermore, it would make visible the need for more interdisciplinarity, counseling, and inter-institutional coordination. These actions would speed up responses to this context of vulnerability and avoid referrals or referrals to the public prosecutor's office. However, the real solution to this situation is the involvement of everyone, as a society, in improving the attention to the EAR.

Finally, it would be pertinent to explore the particular interaction between years of experience as an NCM, gender, and emotions, issues that the present study has failed to satisfy.

Round 2

Reviewer 1 Report

Emotional Universe of Nurse Case managers regarding care for elderly at risk in Spain

Overall, the authors have improved the manuscript. However, there are some issues that have not been improved and they need to be addressed.

1. How many sample sizes were employed in this study?

2. Since the characteristics of the participants vary including gender distribution and NCM’s year of experience, how does the author control such potential confounder variable? Moreover, the author found that gender was also the main issue in the findings.

3. The data collection of the study included the period before and after covid 19. How do the authors conclude the findings since the situations are different?

4. Since the study has been conducted in Andalusia province, could the study findings be generalized to all NCMs across Spain?

Author Response

1.- How many sample sizes were employed in this study?

We apologise if we were unable to inform you in this respect in our previous reply. In this sense, as we informed you in the previous answer, discourse saturation in the pre-pandemic period was reached after analysing the responses of 11 participants. To analyse the period after covid 19, the same 11 participants were contacted. However, the life and contractual changes suffered by 3 of them (NCM4, NCM and NCM PRE2) meant that they were unable to participate in the second moment. In their place, 3 new participants were incorporated. Therefore, in the period after covid 19, the discourses of 11 informants were collected, which was again sufficient to reach discourse saturation

2.- Since the characteristics of the participants vary including gender distribution and NCM’s year of experience, how does the author control such potential confounder variable? Moreover, the author found that gender was also the main issue in the findings.

Rather than being confounding variables, gender and years of experience have been considered, in the study, as criteria of variability/heterogeneity. In other words, it was intentionally "sought" that the results would include the discourses of participants who differed in terms of these characteristics. Based on this intentional heterogeneity, it should be noted that the discourses of the different subjects have been identified (with the corresponding code), so that they can be associated with the characteristics of the subject. Although, as we reported in the previous response: "However, the analysis of the participants' discourse did not show conclusive differences".

3.- The data collection of the study included the period before and after covid 19. How do the authors conclude the findings since the situations are different?

As pointed out in the previous response to the reviewer, the authors' conclusion is that the post-covid-19 situation has only underlined the scenario that was already observed before covid-19: older people at risk with a difficult life situation and professionals facing severe constraints and pressures arising from a precarious social and health care context.

4.- Since the study has been conducted in Andalusia province, could the study findings be generalized to all NCMs across Spain?

In accordance with the characteristics of qualitative research, we cannot ensure the transferability (a priori) of the results to other contexts. However, in order to facilitate such transferability, the authors have provided a detailed and in-depth description of both the contexts and the subjects, so that the audience or reader of the report can determine whether the findings are transferable to their context (on the basis of existing similarities).

Reviewer 3 Report

I state my opinion in the order of the five points that called for revision.

1.Your manuscript would be improved if you could add the following sentences to the abstract section, although there is a limit to the number of words.

“Motivated by the lack of qualitative research in the emotional universe of nurse case managers,”

2.I think the redundancy of the introduction section has already been improved

3.I think the author has indicated what the abbreviation stands for in the figures and tables.

4.I think that readers can understand the contents of the figures even if they only look at Figures 1 and 2.

5.I think the authors discuss further future research in the last paragraph of the conclusion section.

Author Response

First of all, we would like to express our sincere thanks to the reviewer for both the time spent in further revising the manuscript and for his comments. The sentence has been included in the abstract.